eLife | RESEARCH ARTICLE

# Neurexins control the strength and precise timing of glycinergic inhibition in the auditory brainstem

He-Hai Jiang[1,2,3], Ruoxuan Xu[1], Xiupeng Nie[1], Zhenghui Su[1], Xiaoshan Xu[1], Ruiqi Pang[4,5], Yi Zhou[4], Fujun Luo[1,3]*

[1]Guangzhou National Laboratory, Guangzhou, China; [2]Bioland Laboratory, Guangzhou, China; [3]School of Basic Medical Sciences, Guangzhou Medical University, Guangzhou, China; [4]Department of Neurobiology, School of Basic Medicine, Army Medical University, Chongqing, China; [5]Advanced Institute for Brain and Intelligence, School of Medicine, Guangxi University, Nanning, China

*For correspondence: luo_fujun@gzlab.ac.cn

Competing interest: The authors declare that no competing interests exist.

**Abstract** Neurexins play diverse functions as presynaptic organizers in various glutamatergic and GABAergic synapses. However, it remains unknown whether and how neurexins are involved in shaping functional properties of the glycinergic synapses, which mediate prominent inhibition in the brainstem and spinal cord. To address these issues, we examined the role of neurexins in a model glycinergic synapse between the principal neuron in the medial nucleus of the trapezoid body (MNTB) and the principal neuron in the lateral superior olive (LSO) in the auditory brainstem. Combining RNAscope with stereotactic injection of AAV-Cre in the MNTB of neurexin1/2/3 conditional triple knockout mice, we showed that MNTB neurons highly express all isoforms of neurexins although their expression levels vary remarkably. Selective ablation of all neurexins in MNTB neurons not only reduced the amplitude but also altered the kinetics of the glycinergic synaptic transmission at LSO neurons. The synaptic dysfunctions primarily resulted from an impaired $Ca^{2+}$ sensitivity of release and a loosened coupling between voltage-gated $Ca^{2+}$ channels and synaptic vesicles. Together, our current findings demonstrate that neurexins are essential in controlling the strength and temporal precision of the glycinergic synapse, which therefore corroborates the role of neurexins as key presynaptic organizers in all major types of fast chemical synapses.

## eLife assessment

This study provides **important** insights into the role of neurexins as regulators of synaptic strength and timing at the glycinergic synapse between neurons of the medial nucleus of the trapezoid body and the lateral superior olive, key components of the auditory brainstem circuit involved in computing sound source location from differences in the intensity of sounds arriving at the two ears. Through an elegant combination of genetic manipulation, fluorescence in-situ hybridization, ex vivo slice electrophysiology, pharmacology and optogenetics, the authors provide **compelling** and rigorous evidence to support their claims. While further work is needed to reveal the mechanistic basis by which neurexins influence glycinergic neurotransmission, this work will be of interest to both auditory and synaptic neuroscientists.

## Introduction

Neurexins are evolutionarily conserved synaptic adhesion molecules and their genetic mutations are highly associated with autism and schizophrenia (*Gomez et al., 2021*; *Südhof, 2008*). In the

vertebrates, neurexins are encoded by three genes Nrxn1, Nrxn2, and Nrxn3, each of which contains distinct promoters for expressing longer α- and shorter β-neurexins (*Ullrich et al., 1995*; *Ushkaryov et al., 1992*). Both α- and β-neurexins are critical and non-redundant in regulating the formation and function of synapses (*Anderson et al., 2015*; *Missler et al., 2003*; *Südhof, 2017*). Triple knockout (KO) of α-neurexins causes a significant decrease in the density of GABAergic but not glutamatergic synapses while producing a strong reduction in both inhibitory and excitatory synaptic transmission (*Missler et al., 2003*). Triple KO of β-neurexins reveals a significant phenotype only in glutamatergic synapses, but not in GABAergic synapses (*Anderson et al., 2015*). Deletion of all neurexins, including Nrxn1/2/3 α- and β-neurexins, leads to profound but strikingly distinct phenotypes in different synapses, which further points to the diverse actions of neurexins in specific neurons (*Chen et al., 2017*; *Luo et al., 2020*). Consistently, transcriptomic analysis has revealed that the expression profiles of various neurexins differ remarkably in excitatory neurons versus inhibitory neurons even among distinct subtypes of interneurons (*Földy et al., 2016*; *Fuccillo et al., 2015*; *Lukacsovich et al., 2019*; *Uchigashima et al., 2019*). Despite the multifaceted roles of neurexins widely observed in various glutamatergic and GABAergic synapses (*Boxer and Aoto, 2022*; *Chen et al., 2017*; *Luo et al., 2021*; *Südhof, 2017*), however, it remains largely unknown how neurexins may regulate the functional properties of the glycinergic synapses, which mediate essential inhibition in the spinal cord, brainstem and retina (*Assareh et al., 2023*; *Brill et al., 2020*; *Chang et al., 2023*; *Feng et al., 1998*; *Foster et al., 2015*; *Gomeza et al., 2003a*; *Gomeza et al., 2003b*; *Zeilhofer et al., 2012*).

In the mature mammalian auditory brainstem, it is well characterized that the principal neurons in the medial nucleus of the trapezoid body (MNTB) project glycinergic synaptic inputs to the lateral superior olive (LSO) (*Fischer et al., 2019*; *Kotak et al., 1998*; *Nabekura et al., 2004*; *Sanes and Friauf, 2000*). These synapses are robust, precise, and extremely reliable even during high-frequency stimulation (*Krächan et al., 2017*). Strong evidence has shown that the MNTB–LSO glycinergic synapses are subject to extensive refinement during development, including transmitter type switch, the elaboration of dendrites and axonal arbors followed by synaptic pruning, to achieve exquisite control of fast inhibition (*Brill et al., 2020*; *Kim and Kandler, 2003*; *Kotak et al., 1998*; *Nabekura et al., 2004*; *Sanes and Friauf, 2000*). However, the molecular mechanisms of shaping their functional characteristic remain incompletely understood.

We utilized the triple Nrxn1/2/3 conditional KO mice (Nrxn123 cTKO) (*Chen et al., 2017*) to ablate all neurexins in MNTB principal neurons with stereotactic injection of adeno-associated virus (AAV) expressing Cre recombinase. Combining this with the RNAscope technique, we verified that MNTB neurons highly express various isoforms of neurexins. Selective injection of AAV-Cre, but not AAV-EGFP(enhanced green fluorescent protein), successfully blocked expression of all neurexins in MNTB neurons. Functionally, pan-neurexin deletion significantly impaired both the amplitude and the kinetics of the glycinergic synaptic transmission. Surprisingly, deletion of all neurexins led to a significantly higher number of glycinergic synapses innervating the LSO and higher frequency of spontaneous neurotransmitter release. Application of low extracellular $Ca^{2+}$ blocked the glycinergic neurotransmission of both control and neurexin-deficient synapses, but with a significant difference in its effectiveness. Moreover, treatment with EGTA(ethylene glycol tetraacetic acid), a high-affinity $Ca^{2+}$ chelator with slow binding kinetics, caused a much stronger inhibition of neurotransmission in neurexin-deficient synapses, suggesting a significant disruption of the tight coupling between voltage-gated $Ca^{2+}$ channels and synaptic vesicles in the MNTB–LSO glycinergic synapses.

## Results

### Neurexins are highly expressed in MNTB neurons

To study the role and mechanism of neurexins in the glycinergic synapse between MNTB neurons and LSO neurons, which are critical in computing sound localization in mammalian auditory brainstem (*Grothe et al., 2010*), we first characterized the expression of Nrxn1–3 in the principal neurons of the MNTB by fluorescent in situ hybridization (FISH) using the RNAscope technique. As shown in *Figure 1*, MNTB neurons in wild-type (WT) mice expressed Nrxn1, Nrxn2, and Nrxn3, although the expression level of each gene appeared to vary remarkably. As an initial exploration, we took advantage of Nrxn123 cTKO mice that enable deletion of all neurexins following the expression of Cre recombinase (*Chen et al., 2017*), to bypass the functional redundancy of different neurexins.

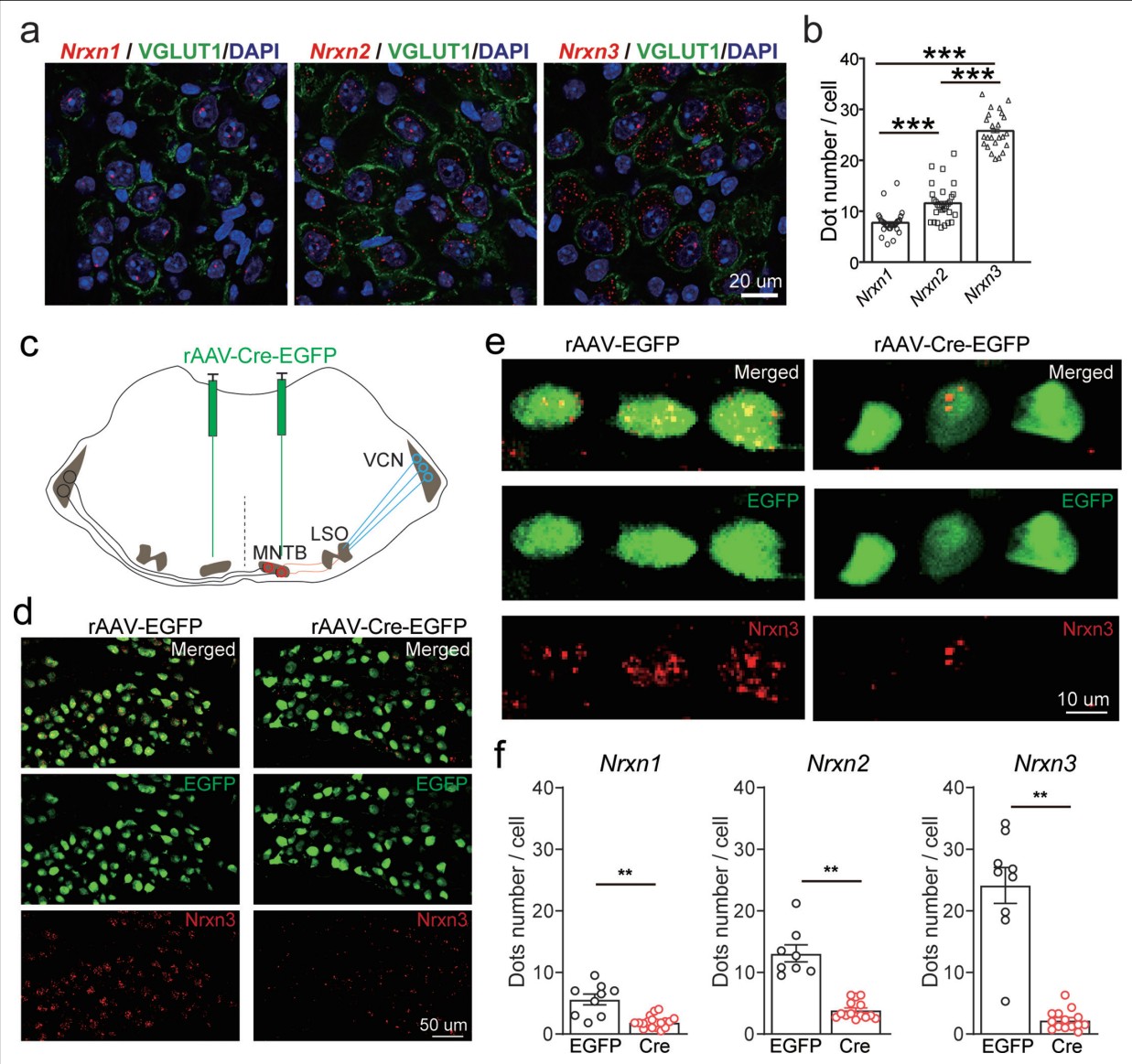

**Figure 1.** Neurexins are highly expressed in medial nucleus of the trapezoid body (MNTB) neurons. (**a**) Immunohistochemistry and RNAscope fluorescent in situ hybridization (FISH) showing expression of various isoforms of neurexins in the MNTB, which make glycinergic synapses with lateral superior olive (LSO) neurons. (**b**) Summary of expression of various isoforms of neurexins in the MNTB of wild-type (WT) mice. Nrxn1 ($n$ = 29), Nrxn2 ($n$ = 29), p = 0.0001; Nrxn1 ($n$ = 29), Nrxn3 ($n$ = 24), p = 0.0001; Nrxn2 ($n$ = 29), Nrxn3 ($n$ = 24), p = 0.0001. (**c**) Diagram of auditory brainstem and virus injection in Nrxn123 cTKO mice. (**d**) Representative images of Nrxn3 FISH in the MNTB of mice injected with AAV-EGFP or AAV-Cre-EGFP. Scale bar: 50 μm. (**e**) Selected ROI(region of interest) in high magnification showing Nrxn3 FISH in MNTB neurons expressing EGFP or Cre-EGFP. Scale bar: 10 μm. (**f**) Summary of expression of various neurexins in the MNTB of Nrxn123 cTKO mice injected with AAV-EGFP and AAV-Cre-EGFP. Nrxn1: EGFP ($n$ = 9), Cre ($n$ = 19), p = 0.0053; Nrxn2: EGFP ($n$ = 8), Cre ($n$ = 14), p = 0.0038; Nrxn3: EGFP ($n$ = 9), Cre ($n$ = 14), p = 0.0016. Data are means ± standard error of the mean (SEM). Number of sections analyzed are indicated in the bars (**b, f**); statistical differences were assessed by Student's $t$-test (**p < 0.01, ***p < 0.001). Source data are provided as a Source Data file.

Neuronal specific deletion of all neurexins was achieved by stereotactic injection of AAV-Cre-EGFP into the MNTB at P0 (TKO). As a control, AAV-EGFP was injected (Ctrl). Successful virus injection and expression of Cre recombinase in MNTB neurons were confirmed by strong EGFP fluorescence (*Figure 1d, e*). FISH data further demonstrated that the recombinase dramatically blocked the expression of all neurexins, as compared to the normal expression of neurexins when EGFP was expressed (*Figure 1f*, compared to *Figure 1b*). Altogether, these data show that the glycinergic neurons in the MNTB highly express various neurexin genes, which can be specifically deleted by stereotactic injection of AAV-Cre in Nrxn123 cTKO mice.

## Neurexins are essential for dictating the strength and kinetics of the MNTB–LSO glycinergic synapse

Since neurexins play an essential role in orchestrating functional organization of various synapses including both glutamatergic and GABAergic synapses, we hypothesize that deletion of all neurexins may have a significant impact on glycinergic synapses. To test the hypothesis, we sought to examine how neurexins may regulate the function of the MNTB–LSO glycinergic synapse in the auditory brainstem. Because LSO neurons receive excitatory glutamatergic inputs and inhibitory GABAergic/glycinergic inputs, we used a pharmacological approach to isolate the glycinergic inputs. By blocking glutamatergic inputs with CNQX and AP5, we were able to selectively record inhibitory postsynaptic currents (IPSCs) from LSO neurons, which can be completely inhibited by strychnine, a potent glycine receptor inhibitor (*Figure 2—figure supplement 1*).

To specifically examine the role of neurexins in regulating the glycinergic synapse, we used an optogenetic approach in which two AAVs, AAV-Cre-EGFP and AAV-DIO-ChR2-mCherry, were co-injected in Nrxn123 cTKO mice. As a control, AAV-ChR2-EYFP was used for injection. Virus injection was performed at P0 and acute slices derived from these mice were analyzed at P13–14. We verified the successful expression of ChR2 by examining the fluorescence and further by recording optogenetically evoked action potentials (APs) from the soma of MNTB neurons. Reliable AP firing can be induced by single or repetitive high-frequency light stimulation (*Figure 2—figure supplement 1c*). In control mice, optogenetic stimulation of MNTB neurons robustly triggered large IPSCs recorded from LSO neurons (*Figure 2a, b*), which can be fully blocked by strychnine (*Figure 2—figure supplement 1d*) and are thus purely glycinergic. In TKO mice with co-injection of Cre and ChR2, however, optogenetic stimulation-induced significantly smaller IPSCs (*Figure 2a, b*). On average, we observed ~60% reduction in the peak amplitude (*Figure 2b*), similar to previous findings at other different synapses (*Chen et al., 2017*; *Luo et al., 2020*). Furthermore, the kinetics of IPSCs were dramatically reduced as indicated by significantly slower rise time and decay time in TKO mice as compared to the control (*Figure 2c, d*). Together, these data strongly suggest that neurexins are essential for the intact function of the glycinergic synapse.

To further address the physiological role of neurexins in AP-evoked neurotransmission at the glycinergic synapse, we applied electric stimulation to activate presynaptic fibers of MNTB neurons by placing a stimulating electrode in the midway between the MNTB and the LSO. Consistent with our optogenetic approach, afferent-evoked large IPSCs were predominantly glycinergic as they can also be fully blocked by strychnine (*Figure 2—figure supplement 1*). A modest but significant decrease in synaptic strength was observed in Nrxn123 TKO mice as compared to control mice (*Figure 2e, f*), confirming the important role of neurexins in the glycinergic synapses. The modest effect of the pan-neurexin deletion on synaptic strength in the MNTB–LSO synapses is likely due to the incomplete AAV transfection in MNTN neurons. Therefore, it is likely that a small fraction of MNTB neurons may still express relatively normal expression of neurexins. It is worth pointing out that extensive efforts were made to optimize our virus injection protocols by adjusting the injection parameters including the needle angle, virus titers, and volume. On average, it was estimated that approximately ~80% neurons in the MNTB expressed GFP and thus were infected by AAV-Cre. It is also possible that other sources of glycinergic inputs other than the MNTB afferents are likely to present but are obscured in WT animals by the predominant glycinergic inputs from MNTB neurons (*Jalabi et al., 2013*).

## The mechanism of neurexins for regulating the MNTB–LSO glycinergic synapse

Since neurexins play multifaceted roles in regulating pre- and/or postsynaptic functions, we explored the potential mechanisms by initially examining the paired-pulse ratio (PPR) of IPSCs, which has been routinely used for identifying a presynaptic origin of regulation. As shown in *Figure 2i*, IPSCs evoked by two consecutive stimuli at various intervals showed a paired-pulse depression in both control and TKO mice. However, the PPRs were significantly increased by deletion of neurexins (*Figure 2j*, *Figure 2—figure supplement 2*), suggesting that neurexins are required for maintaining a high release probability of the glycinergic nerve terminal.

We then recorded and analyzed spontaneous IPSCs from LSO neurons. Compared to the control, there was no significant difference in both the amplitude and kinetics of sIPSCs (*Figure 3a–d*) in TKO mice, indicating the normal content of synaptic vesicle as well as the normal function of postsynaptic

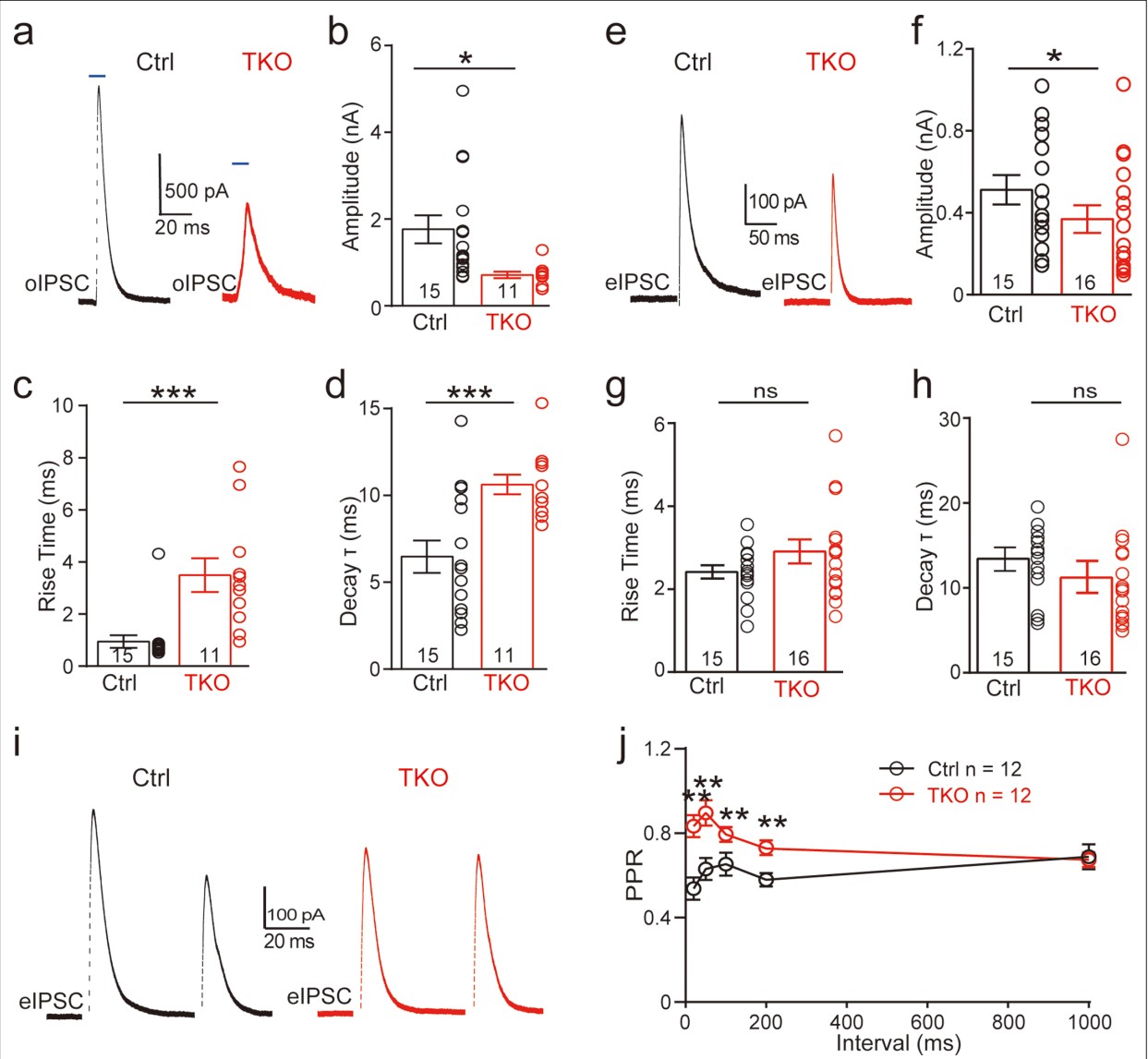

**Figure 2.** Deletion of neurexins reduces the strength and kinetics of glycinergic synapse. (**a**) Representative inhibitory postsynaptic current (IPSC) traces of lateral superior olive (LSO) neurons evoked by optogenetic stimulation of ChR2 expressed in medial nucleus of the trapezoid body (MNTB) neurons. (**b–d**) Summary of IPSC amplitude, rise time, and decay time constant. Ctrl (*n* = 15), TKO (*n* = 11), p = 0.0119, p = 0.0004, p = 0.002. Unpaired two-sided *t*-test. (**e–h**) Same as a–d, except that IPSCs were evoked by afferent fiber stimulation. Ctrl (*n* = 15), TKO (*n* = 16), p = 0.0267, p = 0.152, p = 0.241. (**i**) Representative IPSCs traces of LSO neurons evoked by a pair of fiber stimulation at 50-ms interval. (**j**) Summary of paired-pulse ratio (PPR) of IPSCs in relationship to different paired-pulse intervals. Ctrl (*n* = 12), TKO (*n* = 12), p < 0.0001, two-way analysis of variance (ANOVA). Data are means ± standard error of the mean (SEM). Number of neurons are indicated in the bars (**b–d, f–h**) or graph (**j**). Statistical differences were assessed by Student's *t*-test or two-way ANOVA test (*p < 0.05, **p < 0.01, ***p < 0.001).

The online version of this article includes the following figure supplement(s) for figure 2:

**Figure supplement 1.** The medial nucleus of the trapezoid body (MNTB)–lateral superior olive (LSO) synapse is predominantly glycinergic in mice.

**Figure supplement 2.** Deletion of neurexins increases the paired-pulse ratio (PPR) of optogenetic-evoked inhibitory postsynaptic currents (IPSCs) at the glycinergic synapse.

glycine receptors. Surprisingly, however, we found that deletion of all neurexins caused a small but significant increase in the frequency of sIPSCs (*Figure 3a, b*), which is opposite to previous reports of either no change or decrease in the miniature frequency at glutamatergic or GABAergic synapse (*Chen et al., 2017*; *Luo et al., 2020*). To test whether the increase in sIPSC frequency may be caused by an increase in the number of glycinergic synapses, we performed immunostaining of brainstem

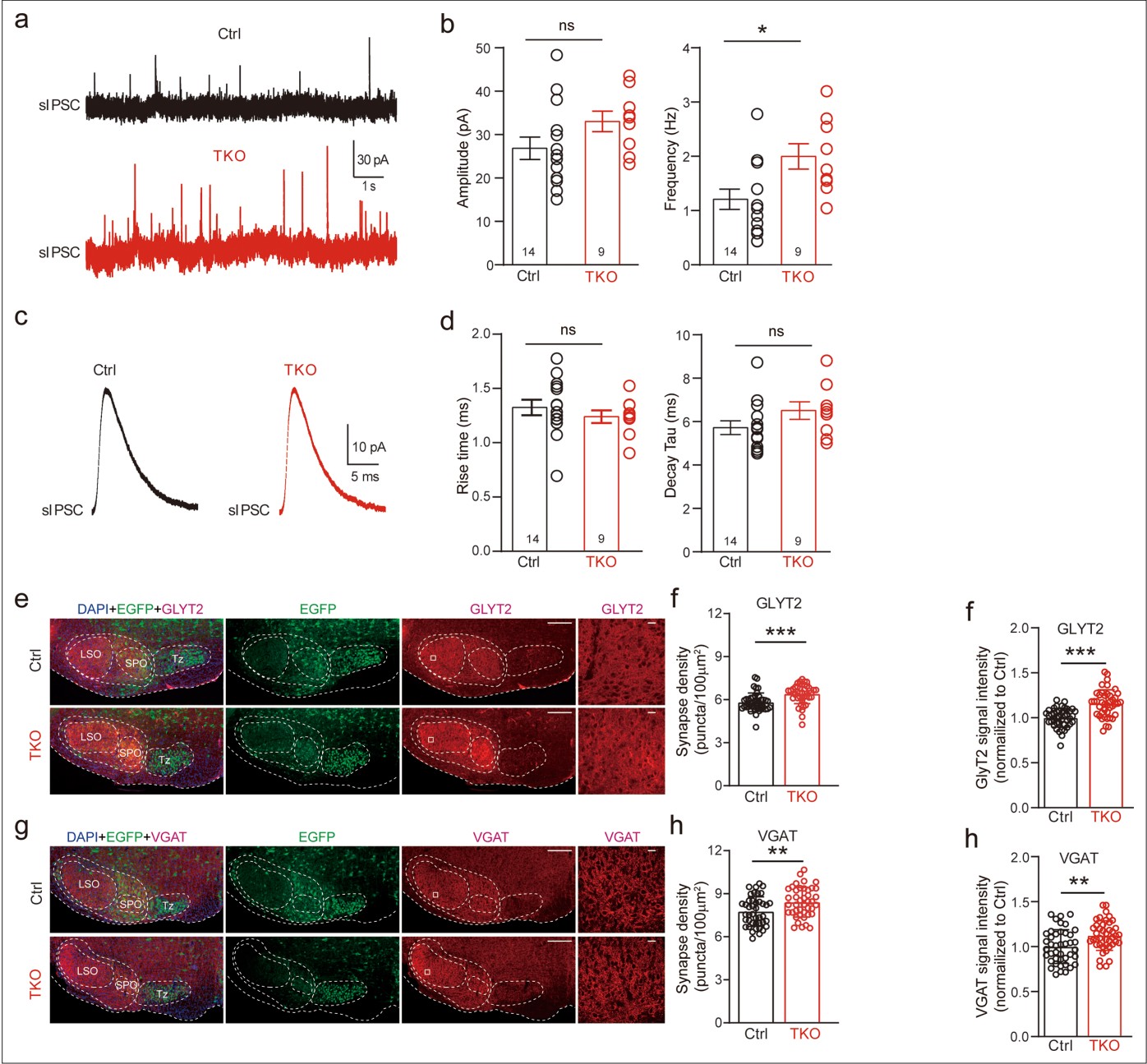

**Figure 3.** Deletion of neurexins increases sIPSC frequency and glycinergic synapse density of lateral superior olive (LSO) neurons. (**a**) Representative sIPSC traces of LSO neurons. (**b**) Summary of sIPSC amplitude and frequency. Ctrl ($n = 14$), TKO ($n = 9$), p = 0.057, p = 0.015, unpaired two-sided $t$-test. (**c**) Representative sIPSC waveforms of LSO neurons. (**d**) Summary of sIPSC rise time and decay time constant. Ctrl ($n = 14$), TKO ($n = 9$), p = 0.376, p = 0.82, unpaired two-sided $t$-test. (**e**) Representative confocal microscopy images of medial nucleus of the trapezoid body (MNTB)–LSO-containing brainstem slice with specific labeling of GlyT2 (red) and EGFP (green) from both control and Nrxn123 TKO mice at P14. Scale bar, 10 μm. (**f**) Summary of glycinergic synaptic density quantified by GlyT2 immunostaining. Ctrl ($n = 42$), TKO ($n = 42$), p = 0.0001, unpaired two-sided $t$-test. (**g, h**) Same as (**e, f**) except for specific labeling of VGAT (red) and EGFP (green). Ctrl ($n = 42$), TKO ($n = 42$), p = 0.0047, unpaired two-sided $t$-test. Data are means ± standard error of the mean (SEM). Number of neurons or sections/animals for immunostaining are indicated in the bars (**b, d, f, h**). Statistical differences were assessed by Student's $t$-test (*p < 0.05, **p < 0.01, ***p < 0.001). Source data are provided as a Source Data file.

sections containing the LSO with antibodies specific for GlyT2 or VGAT(vesicular GABA transporter), synaptic markers for the glycinergic neurons. Consistent with the increased frequency of sIPSCs, we found a significant increase in the density of both GlyT2 (*Figure 3e, f*) and VGAT (*Figure 3g, h*) staining, suggesting more abundant glycinergic terminals at the LSO of TKO mice. Together, these data show that the pan-neurexin deletion in MNTB neurons has surprisingly increased the baseline

glycinergic neurotransmission, which is distinct from the phenotypes observed in glutamatergic and GABAergic neurons (*Chen et al., 2017*; *Luo et al., 2021*).

## Deletion of neurexins impairs the tight coupling between voltage-gated Ca²⁺ channels and synaptic vesicles at the glycinergic synapse

Because neurexins are shown to exert diverse impacts including specifying the function of voltage-gated Ca²⁺ channels and their spatial clustering within the presynaptic active zone (*Chen et al., 2017*; *Luo et al., 2020*; *Missler et al., 2003*), we approached the matter by first exploring the effect of pan-neurexin deletion on the calcium sensitivity of neurotransmitter release at the MNTB–LSO synapse.

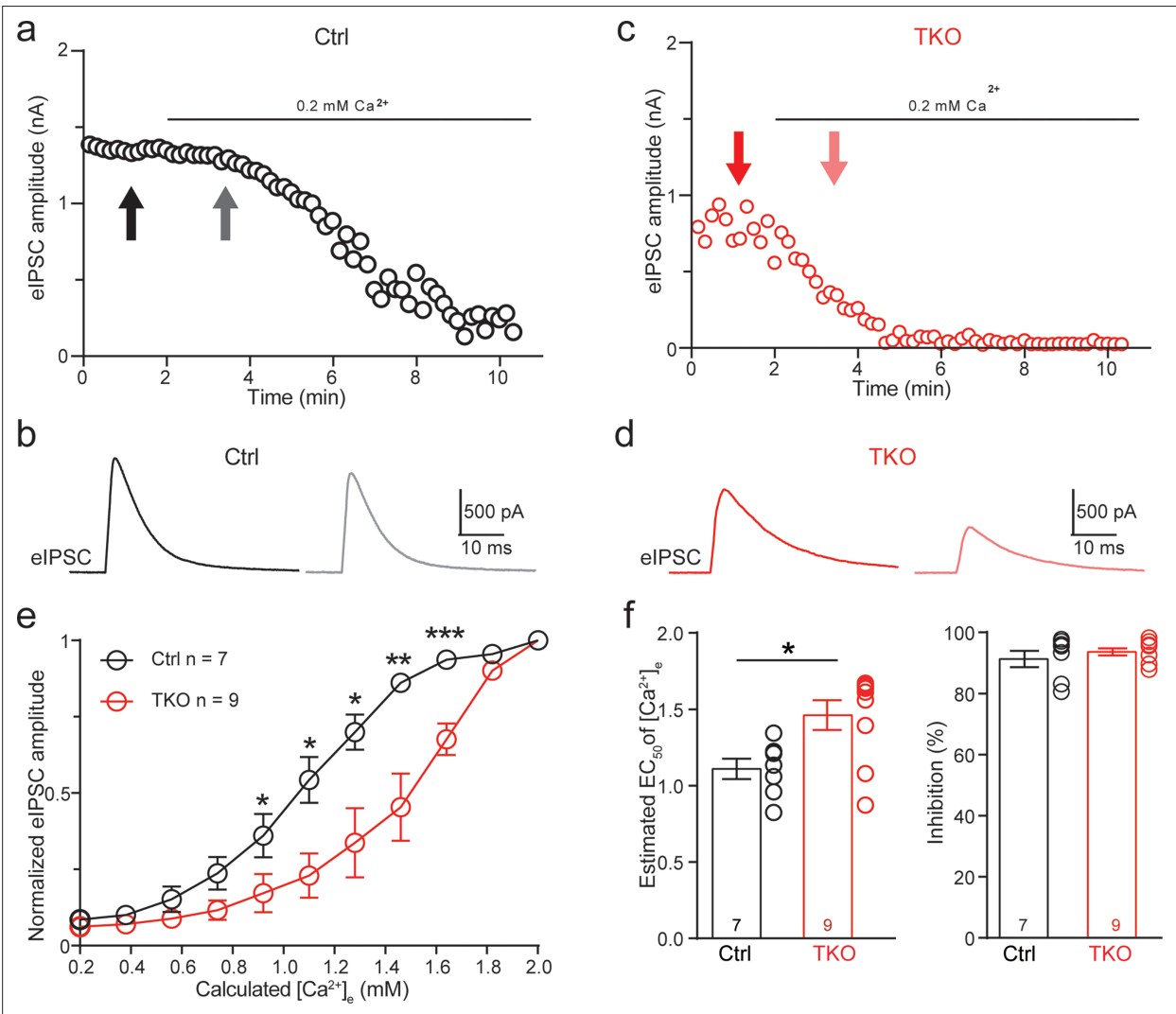

**Figure 4.** Deletion of neurexins increases the Ca²⁺ sensitivity of release at the glycinergic synapse. (**a**) Representative time courses of inhibitory postsynaptic current (IPSC) amplitudes for individual lateral superior olive (LSO) neurons of control mice during the perfusion of low Ca²⁺ ACSF. (**b**) Representative IPSC traces of control mice before and during low Ca²⁺ perfusion at the indicated points in **a**. (**c, d**) Same as (**a, b**), except of TKO mice. (**e**) Summary of normalized IPSC amplitudes plotted against the calculated extracellular Ca²⁺ concentration for control and TKO mice. Ctrl (*n* = 7), TKO (*n* = 9), p = 0.0311 for 0.92 mM Ca²⁺, p = 0.0104 for 1.10 mM Ca²⁺, p = 0.0208 for 1.28 mM Ca²⁺, p = 0.0067 for 1.46 mM Ca²⁺, p = 0.0007 for 1.64 mm Ca²⁺, unpaired two-sided *t*-test. (**f**) Summary of EC₅₀ of Ca²⁺ for half-blocking of IPSCs and the blocking effect of 0.2 mM Ca²⁺ on IPSC amplitude. Ctrl (*n* = 7), TKO (*n* = 9), p = 0.0139, p = 0.3985, unpaired two-sided *t*-test. Data are means ± standard error of the mean (SEM). Number of neurons are indicated in the graph (**e**) or bars (**f**). Statistical differences were assessed by Student's *t*-test (*p < 0.05, **p < 0.01, ***p < 0.001). Source data are provided as a Source Data file.

The online version of this article includes the following figure supplement(s) for figure 4:

**Figure supplement 1.** Pan-neurexin deletion enhances the blocking effects of reducing extracellular Ca²⁺ on glycinergic synaptic transmission.

We perfused the brain slice with low Ca$^{2+}$ (0.2 mM) ACSF(artificial cerebrospinal fluid) while recording IPSCs continuously. In both control and Nrxn123 TKO mice, the perfusion of low Ca$^{2+}$ ACSF gradually decreased the amplitude of IPSCs (*Figure 4a, c*, *Figure 4—figure supplement 1*). However, the reduction rate was much faster in TKO mice such that IPSC amplitudes dropped more than 50% in TKO versus ~10% in control when extracellular [Ca$^{2+}$] reached to approximately 1.5 mM (*Figure 4b, d*). The extracellular [Ca$^{2+}$] blocking 50% of IPSCs were estimated as 1.1 and 1.5 mM, respectively, for control and TKO mice, which are significantly different (*Figure 4e, f*), suggesting that neurexins indeed are crucial for regulating calcium sensitivity of transmitter release at the glycinergic synapse.

One of the critical factors affecting the calcium sensitivity of release is the spatial distance between Ca$^{2+}$ channels and synaptic vesicles (*Eggermann et al., 2011*; *Luo et al., 2020*). Additionally, neurexins have been shown to promote tight coupling of Ca$^{2+}$ channels with synaptic vesicle at the glutamatergic calyx of Held synapse (*Luo et al., 2020*). Because the MNTB–LSO glycinergic synapse is similar on multiple aspects of the functional characteristics, being fast, robust, and precise, to the calyx of Held synapse, we tested whether neurexins are also required for the tight coupling of Ca$^{2+}$ channels with synaptic vesicle. We incubated the brain slice with high-concentration EGTA-AM(Acetoxymethyle ester) and continuously recorded IPSCs from individual LSO neurons. As illustrated in *Figure 5*, EGTA-AM treatment slightly blocked IPSCs at control synapses, suggesting the relative ineffectiveness of EGTA in interfering neurotransmitter release (*Figure 5a, b*, *Figure 5—figure supplement 1*). In contrast, at the neurexin-deficient synapses, EGTA-AM treatment remarkably impaired IPSCs (*Figure 5c, d*, *Figure 5—figure supplement 1*). On average, the blocking effects of EGTA at neurexin-deficient synapses (53.2 ± 5.4%) were much stronger than the effect at control synapses (15.6 ± 4.3%). Therefore, consistent with previous studies at the calyx of Held synapse (*Luo et al., 2020*), our data provide further support that neurexins are key synaptic organizers for tightly clustering voltage-gated Ca$^{2+}$ channels with synaptic vesicles at the glycinergic synapse.

## Discussion

Using the MNTB–LSO synapse as a model, here we provide first assessment of the role of neurexins at glycinergic synapses that mediate prominent synaptic inhibition in the brainstem. Our findings in general strongly corroborate the idea that neurexins are central presynaptic organizers in all major types of fast chemical synapses including glutamatergic, GABAergic, and glycinergic.

Specifically, we have shown that various neurexin genes express concurrently in these glycinergic neurons but the expression level of each appears to vary significantly (*Figure 1*). This finding is consistent with many studies in glutamatergic and GABAergic neurons, which converge to show that multiple genes, isoforms, and splicing sites of neurexins normally co-express in individual neurons and the expression profiles are highly cell-type specific and brain region specific (*Fuccillo et al., 2015*; *Schreiner et al., 2014*; *Treutlein et al., 2014*; *Uchigashima et al., 2019*). Distinct neurexins may interact separately and/or redundantly with diverse signaling molecules to form specific molecular codes for shaping neuronal connectivity and functional properties (*Aoto et al., 2015*; *Aoto et al., 2013*; *Dai et al., 2019*; *Lin et al., 2023*; *Lloyd et al., 2023*; *Südhof, 2017*). The observed differential expression of individual neurexins suggests the possibility that specific isoforms may exert dominant regulatory roles in neurotransmission in different synapses. For instance, compelling evidence indicates that Nrxn3, arguably the best studied neurexin isoform, is selectively required for intact synaptic function in the olfactory bulb GABAergic neurons (*Aoto et al., 2015*; *Trotter et al., 2023*). Understanding the distinct roles of each neurexin isoform in regulating glycinergic synaptic transmission is crucial for future studies. Particularly noteworthy is the potential involvement of Nrxn3, the most abundant isoform, in this synapse.

We further show that neurexins are critical in controlling the strength and precise timing of the glycinergic synaptic inhibition at LSO neurons by clustering Ca$^{2+}$ channels tightly with synaptic vesicles in presynaptic release sites (*Figures 2, 4, and 5*). These data strongly corroborate two perspectives on the function of neurexins as central presynaptic organizers. First, deletion of all neurexins not only weakens the glycinergic synaptic connectivity but also impairs the temporal precision of synaptic inhibition, exactly as it does at the glutamatergic calyx of Held synapse (*Luo et al., 2020*). Since both the glutamatergic calyx of Held synapse and the glycinergic MNTB–LSO synapse in the auditory brainstem are extremely strong and fast (*Beiderbeck et al., 2018*; *Grothe, 2003*), the orchestrating role of neurexins in controlling the timing and magnitude of their synaptic transmission is therefore

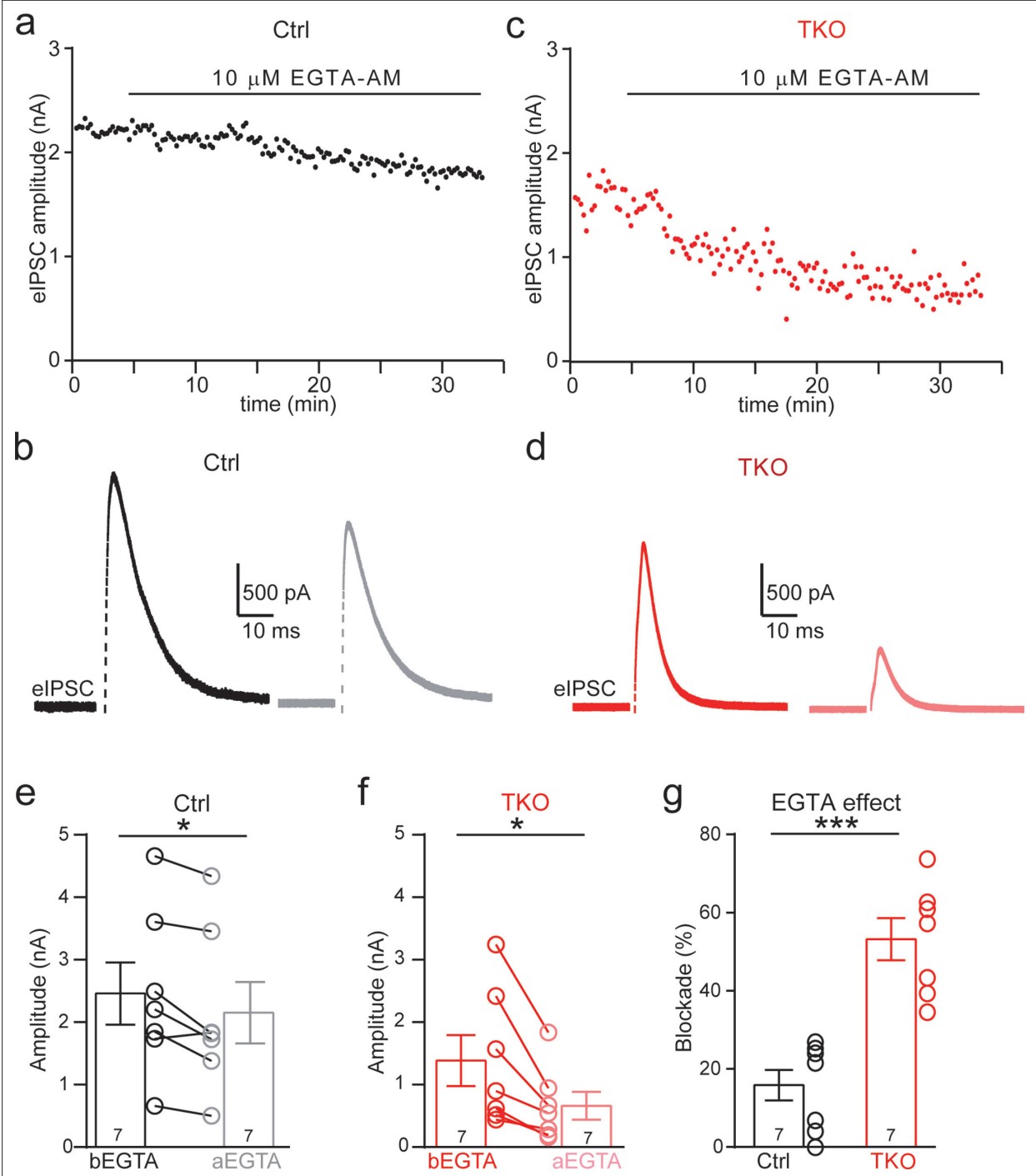

**Figure 5.** Deletion of neurexins increases the blocking effect of EGTA on glycinergic inhibitory postsynaptic currents (IPSCs). (**a**) Representative time courses of IPSC amplitudes for individual lateral superior olive (LSO) neurons of control mice during the treatment of high concentration of EGTA-AM. (**b**) Representative IPSC traces of control mice before and after EGTA. (**c, d**) Same as a, b, except of TKO mice. (**e**) Summary of IPSC amplitude before and after EGTA treatment of control mice. Ctrl (n = 7), TKO (n = 7), p = 0.0201, paired two-sided t-test. (**f**) Summary of IPSC amplitude before and after EGTA treatment of TKO mice. Ctrl (n = 7), TKO (n = 7), p = 0.0123, paired two-sided t-test. (**g**) Summary of the blocking percentage of IPSCs by EGTA. Ctrl (n = 7), TKO (n = 7), p = 0.0001, unpaired two-sided t-test. Data are means ± standard error of the mean (SEM). Number of neurons are indicated in the bars (**e–g**). Statistical differences were assessed by Student's t-test (*p < 0.05, ***p < 0.001). Source data are provided as a Source Data file.

The online version of this article includes the following figure supplement(s) for figure 5:

**Figure supplement 1.** Pan-neurexin deletion enhances the blocking effects of intracellular Ca²⁺ buffer on glycinergic synaptic transmission.

desirable for the auditory information processing. Second, neurexins execute their role at the glycinergic synapse mainly by tightly organizing $Ca^{2+}$ channels within the release sites, again exactly as it does at the calyx of Held synapse (*Luo et al., 2020*). Nanodomain coupling between $Ca^{2+}$ channels and synaptic vesicles has been found in many fast central synapses and is functionally crucial for increasing the efficacy and speed of neurotransmission (*Eggermann et al., 2011*). This similar role of neurexins in clustering $Ca^{2+}$ channels tightly at presynaptic active zones of not only the glutamatergic synapse but also the glycinergic synapse prompts a shared mechanism and further research on the potential molecular candidates.

On the other hand, our data surprisingly reveal that deletion of neurexins may promote the formation/maintenance of the glycinergic synapse, as evidenced by the increase in the frequency of spontaneous synaptic events measured at LSO neurons as well as the increase in the glycinergic synaptic density in Nrxn123 TKO mice (*Figure 3*). Early studies of TKOs of α- or β-neurexins lead to general conclusion that neurexins are not essential for the formation/maintenance of excitatory and inhibitory synapses (*Anderson et al., 2015*; *Missler et al., 2003*). Moreover, deletion of all α- and β-neurexins has no impact on the formation/maintenance of glutamatergic synapse (*Luo et al., 2020*) and GABAergic synapse made by somatostatin-positive interneurons (*Chen et al., 2017*), but produces partial loss (~30%) of GABAergic synapse made by parvalbumin-positive interneurons (*Chen et al., 2017*). There are several possibilities that may explain our present findings on the glycinergic synapse. First, the most parsimonious explanation would be that neurexins play contradictory dual roles at the glycinergic synapse: orchestrate the synchronized synaptic transmission by promoting tight coupling of $Ca^{2+}$ channels with synaptic vesicles at the presynaptic active zone; restrict the formation of weaker synapse or spontaneous transmission. Another well-established example of synaptic protein with dual functions in synaptic transmission is synaptotagmin 1, which acts as a calcium sensor to trigger synchronous release while clamps and restricts spontaneous fusion of vesicles on the other hand (*Südhof, 2013*). It is worthy of note that synaptotagmin can specifically interact with neurexins (*Hata et al., 1993*). Second, deletion of neurexins impairs the strength of the MNTB–LSO synapse, which may indirectly interfere with synaptic elimination process and therefore lead to an increase in synaptic density. Third, the more abundant glycinergic synapses may arise from the developmental compensation of synapse formation by other sources of glycinergic neurons (*Jalabi et al., 2013*). Strong reduction in the strength of the MNTB–LSO synapse caused by selective deletion of neurexins may shift the winner of synaptic competition toward other parallel but much weaker glycinergic inputs in normal animals.

## Methods
### Mouse breeding and virus injection
All animal experiments were approved by the Institutional Animal Care and Use Committee at Bioland Laboratory, Guangzhou National Laboratory, Guangzhou, China (IACUC2020062; GZLAB-AUCP202403A07). Transgenic Nrxn123 cTKO mice (gift from Sudhof's laboratory) (*Chen et al., 2017*) of either sex at ages P13–14 were used for all experiments. Mice were housed at room temperature on a 12-hr light–dark cycle (7:00 to 19:00, light) with food and water freely available.

To delete neurexins in MNTB neurons, we injected P0 mice with AAV2/9-hSyn-EGFP-P2A-Cre-WPRE-hGH-Poly (Cat: PT-0156, BrainVTA Co, China). Mice were anesthetized on ice and immediately transferred to a stereotaxic apparatus (RWD Life Technology, China). Injection was achieved by connecting a glass pipette with tip diameter 20–30 μm to Nanoject III programmable nanoliter injector (Drummond Scientific Company, USA). To enhance virus infection efficiency, we decreased the dosage per injection while increasing the frequency of injections. Additionally, we ensured the pipette remained immobilized for 20–30 s to guarantee virus absorption at injection sites. As a result of this strategy, we estimated that the vast majority of MNTB neurons were inoculated by AAVs. Bilateral injections were performed at the following coordinates: AP from the most rostral point: 5.61 mm, ML: ±0.27 mm, DV: 3.58 mm. For optogenetics experiments, we injected 200 nl (AAV2/9-hSyn-EGFP-P2A-Cre-WPRE-hGH-PolyA and AAV2/9-EF1α-DIO-hChR2(H134R)-mCherry-WPRE-hGH-Poly, 1:1, Cat: PT-0002, BrainVTA Co, China) or 100 nl of AAV2/9-hSyn-hChR2(H123R)-EYFP-WPRE-hGH-PolyA (Cat: PT-1317, BrainVTA Co, China) into each side MNTB.

## Acute slice preparation for electrophysiology

Coronal brain slices containing both the MNTB and LSO were prepared similarly as described previously. In brief, mice of P13–14 were decapitated; brains were rapidly isolated and fixed on the cutting platform a vibratome (VT1200s; Leica, USA), which was immersed in oxygenated cold ACSF containing (in mM): 119 NaCl, 26 NaHCO$_3$, 10 glucose, 1.25 NaH$_2$PO$_4$, 2.5 KCl, 0.05 CaCl$_2$, 3 MgCl$_2$, 2 Na-pyruvate, and 0.5 ascorbic acid, pH 7.4. Transverse slices of 250 µm were sectioned and transferred into a beaker with bubbled ACSF containing (in mM): 119 NaCl, 26 NaHCO$_3$, 10 glucose, 1.25 NaH$_2$PO$_4$, 2.5 KCl, 2 CaCl$_2$, 1 MgCl$_2$, 2 Na-pyruvate, and 0.5 ascorbic acid, pH 7.4. Slices were recovered at 35°C for 30 min, and stored at room temperature (21–23°C) for experiments.

## Whole-cell voltage-clamped recordings

Whole-cell voltage-clamped recordings were made from visually identified LSO neurons in acute brain slices using an Axopatch 700B amplifier (Molecular Devices, USA) and pClamp 11 software (Molecular Devices, USA). Patch pipettes were pulled from borosilicate glass capillaries using a two-stage vertical puller (PC-100, Narashige, Japan). Pipette resistance was 4–6 MΩ when filled with an internal solution containing (in mM): 145 K-gluconate, 6 KCl, 10 HEPES(4-(2-hydroxyethyl)-1-piperazineethanesulfonic acid), 3 Na$_2$-phosphocreatine, 4 Mg-ATP, 0.3 Na$_2$-GTP, 0.2 EGTA, 2 Qx-314; pH 7.2 with KOH. With the extracellular [Cl$^-$] of 133 mM and a temperature of 22°C, the equilibrium potential of the Cl$^-$ was approximately −78.8 mV. In voltage-clamp configuration, IPSCs were recorded by holding cells at 10 mV in ACSF containing CNQX (20 µM), D-AP5 (50 µM), to block AMPA, NMDA receptors, respectively. For some experiments, 2 µM strychnine, a potent glycine receptor antagonist, was added to ACSF to verify that IPSCs recorded at LSO neurons are glycinergic. Membrane potentials were not corrected for liquid junction potential. Electrical fiber stimulation was performed using a concentric bipolar electrode (CBAEB75, FHC Inc, USA) positioned medially to the LSO and MNTB. Short pulses (0.1 ms) were delivered via an isolated stimulator (Model 2100; A-M Systems, USA). For optogenetics, a light simulation of 5 ms was delivered by LED (pE-300 white, CoolLED, UK).

## Whole-cell current-clamp recordings

Whole-cell current-clamped recordings were performed similarly as voltage-clamped recordings, except that the internal solution did not contain Qx-314. APs at MNTB neurons were recorded by holding the membrane potential at approximately −70 mV. For optogenetics, a light simulation of 5 ms was delivered by LED (pE-300 white, CoolLED, UK).

## RNAscope FISH and immunohistochemistry

Mice were anesthetized and perfused at speed of 1 ml/min with 1× phosphate-buffered saline for 10 min followed by 4% paraformaldehyde (PFA) for 10 min. Brains were removed and stored in 4% PFA overnight at 4°C. After going through a 10%, 20%, and 30% sucrose gradient for cryo-protection, the tissue was embedded in Tissue-Tek OCT compound (Cat: 14020108926, Leica) and rapidly frozen on dry ice. Transverse sections at 16 µm were cut at −20°C using a cryostat (CM3050-S, Leica) and mounted directly on Superfrost Plus slides (Cat: PRO-04, Matsunami Platinum Pro). FISH and immunohistochemistry were performed using the RNAscope multiplex platform (Multiplex Reagent Kit, Cat: 323100; Co-detection ancillary Kit, Cat: 323180, Advanced Cell Diagnostics) following the manufacturer's instructions. RNA probes for Nrxn1, Nrxn2, and Nrxn3 (Cat: 461511-C3, 533531-C2, and 505431, respectively, Advanced Cell Diagnostics) and primary antibody against VGluT1 (guinea pig, polyclonal, 1:500, Millipore) were used. Secondary antibodies were Alexa Fluor conjugates (1:400; Thermo Fisher). Samples were mounted with a Vectashield hard-set antifade mounting medium (Cat: H-1500, Vector Laboratories). Images were acquired using Zeiss LSM800 confocal microscope with a ×63 oil-immersion objective (1.4 numerical aperture) or OLYMPUS IXplore SpinSR confocal microscope with a ×60 oil-immersion objective (1.5 numerical aperture). Image analysis was performed using Zeiss 2.6, cellSens Dimension 3.2, and ImageJ.

## Pharmacological manipulations

To estimate the calcium sensitivity of neurotransmitter release, we perfused the slice with ACSF containing low Ca$^{2+}$ (0.2 mM) and continuously recorded eIPSC. EC$_{50}$ of Ca$^{2+}$ sensitivity was determined using a sigmoidal function (Y = Min + (Max − Min)/(1 + 10^((LogEC$_{50}$ − X) ∗ n))), where Y

represents IPSC amplitude, $X$ denotes the calculated extracellular $Ca^{2+}$ concentration, Min refers to the minimal amplitude, and Max represents the maximum amplitude. To estimate the spatial relationship between voltage-gated $Ca^{2+}$ channels and synaptic vesicles, we perfused the slice with ACSF containing 20 μM EGTA-AM for 25–30 min while continuously recording eIPSC. The blocking effect of EGTA on eIPSCs was evaluated by comparing eIPSCs before and after EGTA treatment.

## Statistical analysis

Data were analyzed in the IgorPro (Version 8) and GraphPad Prism 9 software. Summary of data was displayed as mean ± standard error of the mean. Statistical analysis was performed using Student's $t$-test, two-way analysis of variance, and Mann–Whitney $U$-test. Significance of difference was accepted at $p < 0.05$.

## Acknowledgements

We thank Dr. B Zhang (Shenzhen Bay Laboratory) for helpful comments on the manuscript. This study was supported by grants from the National Natural Science Foundation of China (31970914 to FL; 32171001, 32371050 to YZ).

## Additional information

### Funding

| Funder | Grant reference number | Author |
|---|---|---|
| National Natural Science Foundation of China | 31970914 | Fujun Luo |
| National Natural Science Foundation of China | 32171001 | Yi Zhou |
| National Natural Science Foundation of China | 32371050 | Yi Zhou |

The funders had no role in study design, data collection, and interpretation, or the decision to submit the work for publication.

### Author contributions

He-Hai Jiang, Data curation, Formal analysis, Methodology, Writing – review and editing; Ruoxuan Xu, Xiupeng Nie, Zhenghui Su, Data curation, Formal analysis, Methodology; Xiaoshan Xu, Data curation, Formal analysis, Investigation; Ruiqi Pang, Data curation, Methodology; Yi Zhou, Resources, Funding acquisition, Methodology, Writing – review and editing; Fujun Luo, Conceptualization, Supervision, Funding acquisition, Writing - original draft, Writing – review and editing

### Author ORCIDs

Yi Zhou http://orcid.org/0000-0002-2623-8960
Fujun Luo http://orcid.org/0000-0001-9487-6545

Reviewer #1 (Public Review): https://doi.org/10.7554/eLife.94315.3.sa1
Reviewer #2 (Public Review): https://doi.org/10.7554/eLife.94315.3.sa2
Reviewer #3 (Public Review): https://doi.org/10.7554/eLife.94315.3.sa3
Author response https://doi.org/10.7554/eLife.94315.3.sa4

## Additional files

### Supplementary files
• MDAR checklist
• Source data 1. All raw data presented in the article are included in the source data file.

**Data availability**

Source data are provided in *Source data 1*.

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
