## [Editor Report · eLife assessment]

This study provides **important** insights into the role of neurexins as regulators of synaptic strength and timing at the glycinergic synapse between neurons of the medial nucleus of the trapezoid body and the lateral superior olive, key components of the auditory brainstem circuit involved in computing sound source location from differences in the intensity of sounds arriving at the two ears. Through an elegant combination of genetic manipulation, fluorescence in-situ hybridization, ex vivo slice electrophysiology, pharmacology and optogenetics, the authors provide **compelling** and rigorous evidence to support their claims. While further work is needed to reveal the mechanistic basis by which neurexins influence glycinergic neurotransmission, this work will be of interest to both auditory and synaptic neuroscientists.

---

## [Referee Report · Reviewer #1 (Public Review)]

Jiang et al. demonstrated that ablating Neurexins results in alterations to glycinergic transmission and its calcium sensitivity, utilizing a robust experimental system. Specifically, the authors employed rAAV-Cre-EGFP injection around the MNTB in Nrxn1/2/3 triple conditional mice at P0, measuring Glycine receptor-dependent IPSCs from postsynaptic LSO neurons at P13-14. Notably, the authors presented a clear reduction of 60% and 30% in the amplitudes of opto- and electric stimulation-evoked IPSCs, respectively. Additionally, they observed changes in kinetics, alterations in PPR, and sensitivity to lower calcium and the calcium chelator, EGTA, indicating solid evidence for changes in presynaptic properties of glycinergic transmission.

Furthermore, the authors uncovered an unexpected increase in sIPSC frequency without altering amplitude. Although the precise mechanism remains unknown, the authors discussed this complex phenotype by considering various possibilities, including the potential scenario where the augmentation in synapses may result from Nrxn deletion rather than being a causal effect.

---

## [Referee Report · Reviewer #2 (Public Review)]

Summary:

In this manuscript, Jiang et al., explore the role of neurexins at glycinergic MNTB-LSO synapses. The authors utilize elegant and compelling ex vivo slice electrophysiology to assess how the genetic conditional deletion of Nrxns1-3 impacts inhibitory glycinergic synaptic transmission and found that TKO of neurexins reduced electrically and optically evoked IPSC amplitudes, slowed optically evoked IPSC kinetics and reduced presynaptic release probability. The authors use classic approaches including reduced [Ca2+] in ACSF and EGTA chelation to propose that changes in these evoked properties are likely driven by the loss of calcium channel coupling. Intriguingly, while evoked transmission was impaired, the authors reported that spontaneous IPSC frequency was increased, due to an increase in the number of synapses in LSO. Overall, this manuscript provides important insight into the role of neurexins at the glycinergic MNTB-LSO synapse and further emphasizes the need for continued study of both the non-redundant and redundant roles of neurexins.

The authors have addressed all of my previous concerns.

---

## [Referee Report · Reviewer #3 (Public Review)]

Summary:

The authors investigate the hypothesis that neurexins serve a crucial role as regulators of the synaptic strength and timing at the glycinergic synapse between neurons of the medial nucleus of the trapezoid body (MNTB) and the lateral superior olivary complex (LSO). It is worth mentioning that LSO neurons are an integration station of the auditory brainstem circuit displaying high reliability and temporal precision. These features are necessary for computing interaural cues to derive sound source location from comparing the intensities of sounds arriving at the two ears. In this context, the authors' findings build up according to the hypothesis first by displaying that neurexins were expressed in the MNTB at varying levels. They followed this up with deletion of all neurexins in the MNTB through the employment of a triple knock-out (TKO). Using electrophysiological recordings in acute brainstem slices of these TKO mice, they gathered solid evidence for the role of neurexins in synaptic transmission at this glycinergic synapse primarily by ensuring tight coupling of Ca2+ channels and vesicular release sites. Additionally, the authors uncovered a connection between the deletion of neurexins and a higher number of glycinergic synapses of TKO mice, for which they provided evidence in the form of immunostainings and related it to electrophysiological data on spontaneous release. Consequently, this investigation expands our knowledge on the molecular regulation of synaptic transmission at glycinergic synapses, as well as on the auditory processing at the level of the brainstem.

Strengths:

The authors demonstrate substantial results in support of the hypothesis of a critical role of neurexins for regulating glycinergic transmission in the LSO using various techniques. They provide evidence for the expression of neurexins in the MNTB and consecutively successfully generate and characterize the neurexin TKO. For their study on LSO IPSCs the authors transduced MNTB neurons by co-injection of virus carrying Cre and ChR2 and subsequently optogenetically evoke release of glycine. As a result, they observed a significant reduction in amplitude and significantly slower rise and decay times of the IPSCs of the TKO in comparison with control mice in which MNTB neurons were only transduced with ChR2. Furthermore, they observed an increased paired pulse ratio (PPR) of LSO IPSCs in the TKO mice, indicating lower release probability. Elaborating on the hypothesis that neurexins are essential for the coupling of synaptic vesicles to Ca2+ channels, the authors show lowered Ca2+ sensitivity in the TKO mice. Additionally, they reveal convincing evidence for the connection between the increased frequency of spontaneous IPSC and the higher number of glycinergic synapses of the LSO in the TKO mice, revealed by immunolabeling against the glycinergic presynaptic markers GlyT2 or VGAT.

Weaknesses:

A concern is on novelty as this work on the effects of pan-neurexin deletion in a glycinergic synapse is quite consistent with the authors prior work on glutamatergic synapses (Luo et al., 2020).

---

## [Author Response]

The following is the authors’ response to the original reviews.

**eLife assessment**
This study provides important insights into the role of neurexins as regulators of synaptic strength and timing at the glycinergic synapse between neurons of the medial nucleus of the trapezoid body and the lateral superior olive, key components of the auditory brainstem circuit involved in computing sound source location from differences in the intensity of sounds arriving at the two ears. Through an elegant combination of genetic manipulation, fluorescence in-situ hybridization, ex vivo slice electrophysiology, pharmacology, and optogenetics, the authors provide convincing evidence to support their claims. While further work is needed to reveal the mechanistic basis by which neurexins influence glycinergic neurotransmission, this work will be of interest to both auditory and synaptic neuroscientists.

We appreciate the recognition of the significance of our study in shedding light on the role of neurexins in regulating synaptic strength and timing at the glycinergic synapse. Indeed, further investigations are warranted to delve deeper into the specific role of each different variant of neurexins in the future. We hope that our work will spark more interest and collaboration in unraveling the complexities of molecular codes of synaptic function.

**Public Reviews:**

**Reviewer #1 (Public Review):**
Jiang et al. demonstrated that ablating Neurexins results in alterations to glycinergic transmission and its calcium sensitivity, utilizing a robust experimental system. Specifically, the authors employed rAAV-Cre-EGFP injection around the MNTB in Nrxn1/2/3 triple conditional mice at P0, measuring Glycine receptor-dependent IPSCs from postsynaptic LSO neurons at P13-14. Notably, the authors presented a clear reduction of 60% and 30% in the amplitudes of opto- and electric stimulation-evoked IPSCs, respectively. Additionally, they observed changes in kinetics, alterations in PPR, and sensitivity to lower calcium and the calcium chelator, EGTA, indicating solid evidence for changes in presynaptic properties of glycinergic transmission.Furthermore, the authors uncovered an unexpected increase in sIPSC frequency without altering amplitude. Despite the reduction in evoked IPSC, immunostaining revealed an increase in GlyT2 and VGAT in TKO mice, supporting the notion of an increase in synapse number. However, the reviewer expresses caution regarding the authors' conclusion that "glycinergic neurotransmission likely by promoting the synapse formation/maintenance, which is distinct from the phenotypes observed in glutamatergic and GABAergic neurons (Chen et al., 2017; Luo et al., 2021)", as outlined in lines 173-175. The reviewer suggests that this statement may be overstated, pointing out the authors' own discussion in lines 254-265, which acknowledges multiple possibilities, including the potential that the increase in synapses is a consequence rather than a causal effect of Nrxn deletion.

We appreciate the reviewer’s thoughtful evaluation of our study. We agree that our conclusion regarding the promotion of synapse formation/maintenance may have been overstated and recognize the need for a more nuanced interpretation of our findings. Accordingly, we have revised our interpretation by discussing carefully the various possibilities that may cause the observed increase in synapse number in line 256-266.

**Reviewer #2 (Public Review):**
Summary:In this manuscript, Jiang et al., explore the role of neurexins at glycinergic MNTB-LSO synapses. The authors utilize elegant and compelling ex vivo slice electrophysiology to assess how the genetic conditional deletion of Nrxns1-3 impacts inhibitory glycinergic synaptic transmission and found that TKO of neurexins reduced electrically and optically evoked IPSC amplitudes, slowed optically evoked IPSC kinetics and reduced presynaptic release probability. The authors use classic approaches including reduced [Ca2+] in ACSF and EGTA chelation to propose that changes in these evoked properties are likely driven by the loss of calcium channel coupling. Intriguingly, while evoked transmission was impaired, the authors reported that spontaneous IPSC frequency was increased, potentially due to an increased number of synapses in LSO. Overall, this manuscript provides important insight into the role of neurexins at the glycinergic MNTP-LSO synapse and further emphasizes the need for continued study of both the non-redundant and redundant roles of neurexins.

We thank the reviewer for the strong comments and support of our work.

Strengths:This well-written manuscript seamlessly incorporates mouse genetics and elegant ex vivo electrophysiology to identify a role for neurexins in glycinergic transmission at MNTB-LSO synapses. Triple KO of all neurexins reduced the amplitude and timing of evoked glycinergic synaptic transmission. Further, spontaneous IPSC frequency was increased. The evoked synaptic phenotype is likely a result of reduced presynaptic calcium coupling while the spontaneous synaptic phenotype is likely due to increased synapse numbers. While neuroligin-4 has been identified at glycinergic synapses, this study, to the best of my knowledge, is the first to study Nrxn function at these synapses.

We again appreciate the positive feedback on the strengths of our study. We agree that the observed reduction in evoked synaptic transmission and the increase in spontaneous IPSC frequency provide intriguing insights into the function of neurexins in regulating glycinergic synaptic activity.

Weaknesses:The data are compelling and report an intriguing functional phenotype. The role of Neurexins redundantly controls calcium channel coupling has been previously reported. Mechanistic insight would significantly strengthen this study.

We wholeheartedly agree with the reviewer that understanding how neurexins control calcium channel coupling at the presynaptic active zone is crucial for elucidating their role in synaptic transmission. While our current study has provided compelling evidence for the functional phenotypes of pan-neurexin deletion, we recognize the importance of investigating the underlying molecular mechanisms in future research. Exploring these mechanisms would undoubtedly enhance our understanding of neurexin function at various synapses and contribute to advancing the field.

The claim that triple KO of Nrxns from MNTB increases the number of synapses in LSO is not strongly supported.

We agree. Echoing the suggestion made by reviewer 1 (as mentioned above), we acknowledge that the claim regarding the increase in synapse numbers in the LSO following the triple knockout of neurexins from the MNTB was overstated. Consequently, we have revised our conclusions more carefully to reflect this adjustment.

Despite the stated caveats of measuring electrically evoked currents and the more robust synaptic phenotypes observed using optically evoked transmission, the authors rely heavily on electrical stimulation for most measurements.

We acknowledge that optogenetic stimulation offers crucial advantages, and we have provided a balanced discussion of the caveats associated with both methods in our manuscript. Additionally, we have conducted new optogenetic experiments specifically for measuring the paired-pulse ratio in control and Nrxn123 TKO mice. These results have been included as a new supplementary figure (Figure S2).

For experiments involving EGTA and low Ca2+ manipulations, we opted for electrical stimulation due to concerns regarding potential side effects of optogenetics, including the phototoxicity and photobleaching during prolonged light exposure.

The differential expression of individual neurexins might indicate that specific neurexins may dominantly regulate synaptic transmission, however, this possibility is not discussed in detail.

We thank the reviewer for bringing up this important point. The differential expression of individual neurexins indeed suggests that specific neurexins may play dominant roles in regulating synaptic transmission. While our study primarily focused on the collective impact of ablating all neurexins, we acknowledge the significance of exploring the specific contributions of individual neurexin isoforms in the future. Understanding the distinct roles of each neurexin isoform could provide valuable insights into the precise mechanisms underlying synaptic function and plasticity. We have added discussion in our revised manuscript Line223-230.

**Reviewer #3 (Public Review):**
Summary:The authors investigate the hypothesis that neurexins serve a crucial role as regulators of the synaptic strength and timing at the glycinergic synapse between neurons of the medial nucleus of the trapezoid body (MNTB) and the lateral superior olivary complex (LSO). It is worth mentioning that LSO neurons are an integration station of the auditory brainstem circuit displaying high reliability and temporal precision. These features are necessary for computing interaural cues to derive sound source location from comparing the intensities of sounds arriving at the two ears. In this context, the authors' findings build up according to the hypothesis first by displaying that neurexins were expressed in the MNTB at varying levels. They followed this up with the deletion of all neurexins in the MNTB through the employment of a triple knock-out (TKO). Using electrophysiological recordings in acute brainstem slices of these TKO mice, they gathered solid evidence for the role of neurexins in synaptic transmission at this glycinergic synapse primarily by ensuring tight coupling of Ca2+ channels and vesicular release sites. Additionally, the authors uncovered a connection between the deletion of neurexins and a higher number of glycinergic synapses in TKO mice, for which they provided evidence in the form of immunostainings and related it to electrophysiological data on spontaneous release. Consequently, this investigation expands our knowledge on the molecular regulation of synaptic transmission at glycinergic synapses, as well as on the auditory processing at the level of the brainstem.Strengths:The authors demonstrate substantial results in support of the hypothesis of a critical role of neurexins for regulating glycinergic transmission in the LSO using various techniques. They provide evidence for the expression of neurexins in the MNTB and consecutively successfully generate and characterize the neurexin TKO. For their study on LSO IPSCs the authors transduced MNTB neurons by co-injection of virus-carrying Cre and ChR2 and subsequently optogenetically evoke release of glycine. As a result, they observed a significant reduction in amplitude and significantly slower rise and decay times of the IPSCs of the TKO in comparison with control mice in which MNTB neurons were only transduced with ChR2. Furthermore, they observed an increased paired pulse ratio (PPR) of LSO IPSCs in the TKO mice, indicating lower release probability. Elaborating on the hypothesis that neurexins are essential for the coupling of synaptic vesicles to Ca2+ channels, the authors show lowered Ca2+ sensitivity in the TKO mice. Additionally, they reveal convincing evidence for the connection between the increased frequency of spontaneous IPSC and the higher number of glycinergic synapses of the LSO in the TKO mice, revealed by immunolabeling against the glycinergic presynaptic markers GlyT2 or VGAT.

We thank the reviewer for the thoughtful and thorough evaluation of the significance of investigating the role of neurexins in glycinergic transmission at the MNTB-LSO synapse, particularly in the context of auditory processing and sound localization. The positive feedback is greatly appreciated.

Weaknesses:The major concern is novelty as this work on the effects of pan-neurexin deletion in a glycinergic synapse is quite consistent with the authors' prior work on glutamatergic synapses (Luo et al., 2020). The authors might want to further work out novel aspects and strengthen the comparative perspective. Conceptually, the authors might want to be more clear about interpreting the results on the altered dependence of release on voltage-gated Ca2+ influx (Ca2+ sensitivity, coupling).

Regarding the reviewer’s concern about the novelty of our work, we acknowledge that our previous work has explored the effects of pan-neurexin deletion on glutamatergic synapses (Luo et al., 2020). However, we would like to point out that a novelty of our present study indeed stems from the exploration of how different types of synapses converge to employ the same mechanism of synaptic function, particularly in the context of neurexin-mediated regulation. Our previous study focused on glutamatergic synapses, the current study delves into the realm of glycinergic synapses, which represent a distinct population with unique properties and functions. Despite the differences between these synapse types, our findings reveal a commonality in the underlying mechanisms of synaptic regulation mediated by neurexins. This convergence of mechanisms across different synapse types highlights the fundamental role of neurexins in synaptic function and plasticity. By elucidating how neurexins regulate synaptic transmission at both excitatory and inhibitory synapses, we provide valuable insights into the general principles governing synaptic function. In addition, this comparative perspective may shed light on the complex interplay between excitatory and inhibitory neurotransmission, which is crucial for maintaining the balance of neuronal activity and network dynamics.

**Recommendations for the authors:**

**Reviewer #1 (Recommendations For The Authors):**
During the developmental period spanning P3-P12, the MNTB-LSO synapses undergo a transition from GABAergic to glycinergic transmission. It is well-established that Neurexin plays a role in modulating GABAergic transmission. In the authors' experimental system, AAV was injected at P0, likely impacting GABAergic transmission, including potentially influencing synapse number, before subsequently affecting glycinergic transmission. A thoughtful discussion of how the experimental interventions might have influenced this developmental process and glycinergic transmission would enhance the clarity and interpretation of their findings.

We thank the reviewer for raising the interesting topic of the transmitter switch during neurodevelopment. Strong evidence using gerbils and rats as animal models demonstrates that the MNTB-LSO synapses undergo a shift from GABAergic to glycinergic during the early development. However, in a more recent study by Friauf and colleagues (Fisher et al., 2019), patch-clamp recordings in acute mouse brainstem slices at P4-P11 combined with pharmacological blockade of GABAA receptors and/or glycine receptors clearly demonstrated no GABAergic synaptic component on LSO principal neurons, suggesting the transmitter subtype switch may be species different. We add a discussion in our revision to clarify this topic.

**Reviewer #2 (Recommendations For The Authors):**
The data are compelling and report an intriguing functional phenotype. Mechanistic insight into how this phenotype manifests would significantly strengthen this study. For example, which neuroligin is found at these MNTB-LSO synapses?

We agree that investigating the underlying molecular mechanisms, particularly the specific function of each variant of neurexins and their respective ligands on the postsynaptic neurons, is crucial. Exploring these mechanisms, which extend beyond the scope of our current study, would undoubtedly enhance our understanding of neurexin function at various synapses and foster advancements in the field.

Does the TKO alter the ability of MNTB inputs to induce AP firing in LSO neurons?

Activation of the MNTB inputs does not directly induce AP firing in LSO neurons, because the MNTB-LSO synapses are glycinergic and serve to inhibit neuronal activity.

We think the reviewer was to ask whether pan-neurexin deletion in the MNTB neurons alter their ability to impact the firing of LSO neurons. Indeed, the weakening of glycinergic transmission due to pan-neurexin ablation in MNTB neurons could potentially alter the excitation-inhibition (E/I) balance, thereby impacting the overall excitability of LSO neurons. We have conducted preliminary experiments to investigate this aspect and found that the E/I balance at LSO neurons was notably increased in TKO mice. We are currently preparing a manuscript to comprehensively address the role of neurexins at the auditory circuit and behavior levels.

Additional calcium measurements using GECIs would provide insight into whether nanodomain calcium or total calcium is altered at these synapses.

We appreciate the valuable suggestion provided by the reviewer. However, distinguishing between Ca2+ nanodomain and Ca2+ microdomain using Ca2+ imaging techniques requires advanced systems such as two-photon STED microscopy, which are beyond the scope of our current research.

It is unclear why fluorescence intensity is quantified instead of the number of synaptic clusters in LSO. In addition to changes in synapse numbers, fluorescent intensity can indicate a number of other possible morphological changes.

We appreciate the valuable suggestion from the reviewer. We have re-analyzed our imaging data to compare synaptic density. The results, as included in Fig.3f and 3h, confirm an increase in the number of glycinergic synapses after pan-neurexin deletion.

The most robust synaptic phenotypes were produced by measuring light-evoked oIPSCs and the authors acknowledge that electrically-evoked eIPSCs might be contaminated by uninfected fibers or by other sources of glycinergic inputs. I suggest that IPSC PPRs, EGTA, and low Ca2+ experiments be performed using optogenetics.

As discussed in our response to Public Reviews, we acknowledge that optogenetic stimulation offers crucial advantages, and we have provided a balanced discussion of the caveats associated with both methods in our manuscript. Additionally, following the reviewer’s suggestion, we have conducted new optogenetic experiments specifically for measuring the paired-pulse ratio in control and Nrxn123 TKO mice. We included this new dataset in supplementary Figure S2, which is consistent with our result obtained with electrically fiber stimulation.

For experiments involving EGTA and low Ca2+ manipulations, we opted for electrical stimulation due to major concerns regarding potential side effects of optogenetics, including the phototoxicity and photobleaching during prolonged light exposure.

It is sometimes confusing which type of evoked stimulation is being used (e.g. PPR, EGTA, and low Ca2+ experiments). To aid in the interpretations of these experiments, it would help to clarify.

We appreciate the reviewer's suggestion regarding the clarity of the evoked stimulation methods used in our experiments. We have revised the manuscript to provide clearer descriptions of the specific types of evoked stimulation employed in each experiment. Thank you for guiding towards this clarification.

The comparisons to Chen et al 2017 and the senior author's 2020 paper seem disjointed and do not contribute to the findings, which alone, are quite interesting. Given the prevailing notion that neurexins control different synaptic properties depending on the brain region and/or synapse studied, is it surprising that the findings observed here differ from previous studies of different synapses (glutamatergic and GABAergic)?

By comparing previous studies at different types of neurons/synapses, our findings reveal a commonality in the underlying mechanisms of synaptic regulation mediated by neurexins. This convergence of mechanisms across different synapse types highlights the fundamental role of neurexins in synaptic function and plasticity. In addition, this comparative perspective may shed light on the complex interplay between excitatory and inhibitory neurotransmission, which is crucial for maintaining the balance of neuronal activity and network dynamics.

Despite Nrxn3 being the most abundant Nrxn mRNA in MNTB neurons, the possible contributions of this highly expressed protein are not discussed.

We thank the reviewer for bringing up this important point. The differential expression of individual neurexins indeed suggests that specific neurexins may play dominant roles in regulating synaptic transmission. While our study primarily focused on the collective impact of ablating all neurexins, we acknowledge the significance of exploring the specific contributions of individual neurexin isoforms in the future. Understanding the distinct roles of each neurexin isoform could provide valuable insights into the precise mechanisms underlying synaptic function and plasticity. We have added discussion in our revised manuscript Line223-230.

**Reviewer #3 (Recommendations For The Authors):**
• There are several instances of spaces missing and typos, please carefully check the manuscript.

We greatly appreciate the reviewer's helpful feedback on the text that could be clarified or improved. We have meticulously edited the manuscript to address these concerns.

• While studying the properties of IPSC, apart from optogenetic stimulation, the authors performed experiments with electrical fiber stimulation. Their findings showed a slightly significant reduction of the IPSC amplitude and no effect on the IPSCs kinetics when comparing the TKO and control. One weakness is the discrepancy between the results from the optogenetic and fiber stimulation experiments, which the authors contribute to inefficient transfection in the fiber stimulation experiments. The authors state that they tried to optimize their protocols for virus injection protocols. However, they do not elaborate on how the transfection rates could be improved in the discussion section. Moreover, it would be good to further address the reasons for the difference in amplitude between the control IPSCs in the optogenetic and fiber stimulation experiments.

Echoing the suggestion by Reviewer 2 (see above), we acknowledge that optogenetic stimulation offers certain advantages, and we have provided a balanced discussion of the caveats associated with both methods in our manuscript. In addition, we have performed a new set of optogenetic experiment for the paired-pulse ratio measurement in control and Nrxn123 TKO mice and included as a new figure in supplementary figure S2.

For experiments involving EGTA and low Ca2+ manipulations, we opted for electrical stimulation due to major concerns regarding potential side effects of optogenetics, including the phototoxicity and photobleaching during prolonged light exposure.

We added the detail of virus injection strategy that optimized the transfection rates in the method section “To enhance virus infection efficiency, we decreased the dosage per injection while increasing the frequency of injections. Additionally, we ensured the pipette remained immobilized for 20-30 seconds to guarantee virus absorption at injection sites. As a result of this strategy, we estimated that the vast majority of MNTB neurons were inoculated by AAVs.” See line288-290.

• Abstract: "ablation of all neurexins in MNTB neurons reduced not only the amplitude but also altered the kinetics of the glycinergic synaptic transmission at LSO neurons."

Changed as suggested.

• Consider revising to "The synaptic dysfunctions primarily resulted from an altered dependence of release on voltage-gated Ca2+ influx."

We appreciate the reviewer's suggestion, which helps improve the clarity of our manuscript. We have revise the phrasing as follows: "The synaptic dysfunctions primarily resulted from an impaired calcium sensitivity of release and a loosened coupling between voltage-gated calcium channels and synaptic vesicles."

• Line 39 should be vertebrates.

Revised as suggested.

• Line 49 it would sound better to say "which further points to the diverse actions of neurexins in specific neurons."

Revised as suggested.

• Line 60 - this paragraph could include information about GABA signaling from the MNTB to the LSO, because on line 113 you mention LSO neurons receive inhibitory GABAergic/glycinergic inputs, but when you do not mention blocking of GABA currents to isolate the glycinergic ones.

We thank the reviewer for the thoughtful and detailed suggestion. We revised the text in line 60 to “In the mature mammalian auditory brainstem” and in line 113, we removed GABAergic to emphasize the nature of glycinergic synapse, particularly in the mouse brainstem where no GABAergic components are found (Fisher et al., 2019).

• Line 72/73 it should be adeno-associated virus; line 73: "combining this with the RNAScope technique" sounds better.

Changed as suggested.

• Line 91 using the RNAScope technique; lines 97, 119 as a control; line 108 the functional organization.

Changed as suggested.

• Line 113 should be a pharmacological approach; line 122 optogenetically evoked.

Changed as suggested.

• Line 132, 160: the control.

Changed as suggested.

• Line 147 thus were infected; line 148 likely to be present but were obscured .

Changed as suggested.

• Line 154 which has been routinely used.

Changed as suggested.

• Line 155 It is not supposed to be Figure 2h but 2i; following that Figure 2i should be 2j; in my opinion, Figure 2i does not display a strong depression for the TKO mice.

Changed as suggested.

• Line 171 a better flow is achieved by saying: together these data show.

Changed as suggested.

• EC50 rather than IC50 of [Ca2+].

Changed as suggested.

• 180 it is better to say "we approached the matter by..."; line 183 while recording;

Changed as suggested.

• Line 203 were much stronger than the effect at control synapses; line 206 tightly clustering.

Changed as suggested.

• Line 212 sounds like they provide evidence for retina and spinal cord as well, should be made clear.

Changed as suggested.

• Line 289 previously.

Changed as suggested.

• Line 295 should be 30 min.

Changed as suggested.

• Line 336, 337 confocal microscope.

Changed as suggested.

• Please provide the number of data points also in figure captions or in the results section.

Added in the captions as suggested.

• Line 533, a better phrasing would be: the blocking effect of 0.2 mM Ca on IPSC amplitude.

Changed as suggested.

• Explain either in the methods or result section how was the EC50 of Ca2+ calculated.

Added in the methods as suggested.